

# *ybx1* acts upstream of *atoh1a* to promote the rapid regeneration of hair cells in zebrafish lateral-line neuromasts

Caleb C. Reagor[1,2,3], Paloma Bravo[1] and A.J. Hudspeth[1]

[1] Howard Hughes Medical Institute and Laboratory of Sensory Neuroscience, The Rockefeller University, New York, NY, United States of America
[2] Department of Physiology and Pharmacology, Karolinska Institutet, Stockholm, Sweden
[3] Weill Medical College, Cornell University, New York, NY, United States of America

## ABSTRACT

Like the sensory organs of the human inner ear, the lateral-line neuromasts (NMs) of fish such as the zebrafish (*Danio rerio*) contain mechanosensory hair cells (HCs) that are surrounded by supporting cells (SCs). A damaged NM can quickly regenerate new HCs by expressing genes such as *atoh1a*, the master regulator of HC fate, in the SCs at the NM's center. We used the supervised learning algorithm DELAY to infer the early gene-regulatory network for regenerating central SCs and HCs and identified adaptations that promote the rapid regeneration of lateral-line HCs in larval zebrafish. The top hub in the network, *Y-box binding protein 1* (*ybx1*), is highly expressed in HC progenitors and young HCs and its protein can recognize DNA-binding motifs in cyprinids' candidate regeneration-responsive promoter element for *atoh1a*. We showed that NMs from *ybx1* mutant zebrafish larvae display consistent, regeneration-specific deficits in HC number and initiate both HC regeneration and *atoh1a* expression 20% slower than in wild-type siblings. By demonstrating that *ybx1* promotes rapid HC regeneration through early *atoh1a* upregulation, the results support DELAY's ability to identify key temporal regulators of gene expression.

## INTRODUCTION

Unlike humans and other mammals, fishes and amphibians can regenerate damaged receptors in the sensory organs of their inner ears (*Corwin, 1981*; *Lombarte et al., 1993*; *Baird, Torres & Schuff, 1993*; *Monroe, Rajadinakaran & Smith, 2015*). Because these aquatic species can also regenerate the related mechanoreceptors in their lateral-line organs for sensing "touch at a distance," they are attractive models for studying conserved and species-specific processes of sensory-cell regeneration (*Jones & Corwin, 1993*; *Cruz et al., 2015*). During development, the lateral lines form when cranial placode-derived cells migrate through the skin and deposit sensory units known as neuromasts (NMs) along the head, trunk, and tail (*Metcalfe, 1985*; *Gompel et al., 2001*; *Chitnis, Dalle Nogare & Matsuda, 2012*; *Nogare et al., 2017*). Like inner-ear sensory epithelia, NMs contain two principal cell types: mechanosensory hair cells (HCs) for detecting directional displacement of water

Corresponding author
Caleb C. Reagor,
creagor@rockefeller.edu

and supporting cells (SCs) that surround the HCs and can serve as HC progenitors during development and regeneration (*Balak, Corwin & Jones, 1990*; *Itoh & Chitnis, 2001*; *Kindt, Finch & Nicolson, 2012*; *Lush et al., 2019*). Fish such as the zebrafish (*Danio rerio*) rely on sensory information from their lateral lines to perform important behaviors such as rheotaxis and escape response (*Montgomery, Baker & Carton, 1997*; *McHenry et al., 2009*). However, due to their location near the surface of the skin, NMs often receive chemical and mechanical insults that damage HCs and leave larvae and adults vulnerable to predation (*Smith & Monroe, 2016*).

Following damage, NMs can regenerate new HC progenitors from competent SCs in 3–5 h to produce new mature HCs in less than 10 h (*Harris et al., 2003*; *Baek et al., 2022*). NMs are accordingly among the fastest-regenerating vertebrate sensory organs. After recruiting macrophages to clear away debris from damaged cells, NMs initiate proliferation in the SCs of their dorsal and ventral regions, each of which undergoes a symmetric division to form two daughter SCs (*Romero-Carvajal et al., 2015*; *Carrillo et al., 2016*; *Thomas & Raible, 2019*; *Denans et al., 2022*). However, some of the SCs located near the NM's center can instead upregulate expression of the transcription factor (TF) *atoh1a*—the master regulator of HC fate—and differentiate into unipotent HC progenitors, subsequently giving rise to pairs of daughter HCs (*Bermingham et al., 1999*; *Sarrazin et al., 2006*; *Cai et al., 2013*; *Chonko et al., 2013*; *Jiang et al., 2014*). The ability of NMs to quickly replace damaged HCs therefore depends on their maintenance of these central SCs and the prompt induction of key HC-specific genes such as *atoh1a*, *pou4f3*, and *gfi1* (*Yu, Wang & Chen, 2020*; *Yu et al., 2021*). The expression timing of SC-specific factors such as *sox2* is also important because deletion of that gene's upstream regeneration-responsive enhancer impedes HC regeneration as well (*Jimenez et al., 2022*).

Because NMs' capacity to regenerate greatly depends on their ability to regulate genes' expression timing, recent studies have employed single-cell RNA-sequencing (scRNA-seq) to examine NMs' dynamic transcriptional changes following damage (*Baek et al., 2022*). These studies can facilitate the reconstruction of dynamic gene-regulatory networks (GRNs) such as those driving the rapid regeneration of HCs and SCs. Our group recently developed DELAY, a convolutional neural network for the identification of direct, causal interactions between TFs and their target genes from pseudotime-ordered single-cell gene expression trajectories (*Reagor, Velez-Angel & Hudspeth, 2023*). DELAY employs transfer learning to reconstruct GRNs and can identify key temporal hubs such as *ESRRB* during the differentiation of Purkinje neurons (*Mannens et al., 2024*). Unlike many methods, DELAY employs ground-truth information from chromatin immunoprecipitation with sequencing (ChIP-seq) or assay of transposase-accessible chromatin using sequencing (ATAC-seq) to train data set-specific models whose predictions are tailored to specific biological contexts. Here, we trained new models of DELAY to reconstruct the dynamic GRNs across regenerating central SCs and HCs and identified novel gene-regulatory adaptations that fostered the rapid regeneration of damaged NMs in larval zebrafish. These mechanisms might present novel strategies for promoting HC regeneration in humans as well (*Iyer & Groves, 2021*). Portions of the following text were previously published as part of a preprint (*Reagor & Hudspeth, 2024*).
## MATERIALS AND METHODS

### Pre-processing of single-cell RNA-sequencing data and pseudotime inference

We used scRNA-seq data from (*Baek et al., 2022*) of neomycin-treated NMs at 0, 0.5, 1, 3, 5, and 10 h after HC ablation and employed the same pre-processing steps in Seurat (version 5.0.1) as in the original study (*Hao et al., 2024*). In brief, cells were filtered according to previous quality control thresholds (300 <number of expressed genes <6,000; mitochondrial counts <5%), followed by count- and log-normalization (Seurat::NormalizeData), and variable feature selection (Seurat::FindVariableFeatures; number of features = 5,000). Samples from individual timepoints were then integrated using reciprocal PCA (RPCA; number of integration features = 3,500). Dimensionality reduction was performed on the integrated data using PCA (number of dimensions = 30) followed by $t$-SNE (number of basis PCs = 10). Cells were then labeled as either HC progenitors, young HCs, mature HCs, central SCs, dorsoventral SCs, anteroposterior SCs, amplifying SCs or mantle cells according to their expression of known marker genes in the computed Louvain clusters (number of basis PCs = 10; resolution = 1.7).

Prior to pseudotime inference, we subset separate regeneration trajectories for central SCs (0–10 hpa) and HCs (central SCs, 0–3 hpa; HC progenitors, 1–5 hpa; young and mature HCs, 3–10 hpa). Due to their strong similarities across all timepoints, we used the original, unintegrated data to compute a new $t$-SNE embedding for the central SCs. For the HC trajectory, we instead used the RPCA-integrated data to compute the updated $t$-SNE embedding. We used Slingshot (version 2.8.0) with clusters from the original timepoints or assigned cell types to infer cells' unique pseudotime values across the trajectories for SCs and HCs, respectively (*Street et al., 2018*).

### Transfer learning and GRN inference with DELAY

Following scRNA-seq pre-processing and pseudotime inference, we used DELAY (version 0.2.0) to infer the HC- and SC-specific GRNs during early regeneration (*Reagor, Velez-Angel & Hudspeth, 2023*). We trained DELAY's RNA-specific model jointly on ground-truth interactions for both HC and SC trajectories using published targets of *sox2*, *sox21a*, *six1a*, and *six1b* from scATAC-seq of inner-ear HC regeneration (*Jimenez et al., 2022*). For both trajectories, we used targets of *sox2* and *six1b* to train the model. For validation, we held out the targets of *six1a* and *sox21a* for the HCs and SCs, respectively. The complete GRNs for 303 TFs expressed in either lineage were then inferred independently for each trajectory. To construct a single GRN for early regeneration, we combined DELAY's maximum predictions per TF-target pair and selected the top twenty regulators for each gene. This approach produced a realistic network containing an approximately scale-free outdegree distribution while limiting genes' in-degrees to reasonable values for future *in silico* simulations.

### *De novo* discovery and relative enrichment of Ybx1's bipartite motif

To identify enriched motifs in *ybx1*'s predicted targets, we used twoBitToFa to gather the enhancer sequences spanning 50 kb upstream and downstream of the transcription start

sites for each TF in the combined GRN for early regeneration (UCSC Command Line Utilities; UCSC zebrafish genome GRCz11). We then divided the sequences into 100 bp fragments, which we classified as either controls or predicted *ybx1* targets. To identify enriched motifs in the predicted targets' enhancers relative to those of the controls, we performed discriminative motif analysis using STREME (version 5.5.5) (*Bailey, 2021*). We then used FIMO (version 5.5.5) to identify individual occurrences of Ybx1's bipartite motif across the enhancers of the predicted targets and controls (*Grant, Bailey & Noble, 2011*). We computed the relative enrichment of Ybx1's bipartite motif occurrences across the promoters of the predicted targets *versus* the controls by first mean-normalizing then smoothing (rolling window = 50 bp) the binned motif occurrences (width = 10 bp) across the enhancers. We then performed min-max scaling of the empirical probability distributions and defined the relative enrichment as the gain in the normalized frequency of motif occurrences per bin, *i.e.,* f (*ybx1* target occurrences)/f (control occurrences).

## Conservation of *atoh1a*'s candidate regeneration-responsive promoter element

The conservation of *atoh1a*'s candidate regeneration-responsive promoter element was determined using a simple alignment of that gene's orthologues across *Danio rerio* (GRCz11 [ensembl]), *Cyprinus carpio* (Cypcar_WagV4.0 [ensembl]), *Carassius auratus* (ASM336829v1 [ensembl]), *Labeo rohita* (IGBB_LRoh.1.0 [NCBI]), *Lepisosteus oculatus* (LepOcu1 [ensembl]), *Mus musculus* (GRCm39 [ensembl]), and *Homo sapiens* (GRCh38.p14 [ensembl]).

## Zebrafish husbandry, strains, genotyping, and HC ablations

Experiments were performed on zebrafish larvae 3–5 dpf in accordance with the standards of Rockefeller University's Institutional Animal Care and Use Committee (protocol number 22,081 H [PRV 19,076 H]). Eggs were collected and maintained at 28.5 °C in egg water containing 1 mg/L methylene blue. Heterozygous *ybx1$^{sa34489}$* zebrafish larvae were obtained from the Zebrafish International Resource Center. The following primers were used for PCR and Sanger sequencing of *ybx1* mutant offspring: 5′-GTGGAGAGATGTGACAGAATATCG-3′ (forward); 5′-CATAACTGAAATAAACCCT GGAGCG-3′ (reverse). We also used *Tg(atoh1a:dTomato)* larvae to visualize newly regenerated HCs by fluorescence microscopy.

To ablate the first primordium-derived HCs, we treated 3 dpf larvae twice for 1 h each with 300 µM neomycin sulfate (Sigma-Aldrich, St. Louis, MO, USA) with 6 h of recovery between treatments. We performed the second treatment to kill all non-regenerating HCs that survived or matured after the first treatment. Though their HCs are somewhat less sensitive to ablation, we used 3 dpf larvae because their lateral-line NMs contained fewer HCs and were therefore more susceptible to synchronization of HC regeneration timing post-ablation.

### Fixation, immunolabeling, and fluorescence imaging of zebrafish larvae

To label lateral-line HCs, we fixed 3–5 dpf zebrafish trunks with 4% formaldehyde in phosphate-buffered saline (PBS) solution with 0.1% Tween-20 (0.1% PBST) for either 1 h at room temperature or overnight at 4 °C. For end-point experiments with inbred *ybx1* mutant offspring, we washed the trunks for 30 m in fresh 0.1% PBST, followed by a 1 h incubation at room temperature in 0.05% PBST with DAPI (1:200) and Alexa Fluor Plus 405 phalloidin (1:40; Invitrogen, Eugene, USA). For time-course experiments with *Tg(atoh1a:dTomato)* larvae, we washed the trunks in 0.5% PBST and incubated them for 1 h at room temperature in 0.05% PBST blocking solution with 1% BSA and again for 2 h in fresh blocking solution with DAPI (1:200), phalloidin (1:40) and goat anti-tdTomato DyLight550-conjugated antibodies (1:50; MyBioSource, San Diego, CA, USA). We washed the trunks once more for 20–30 m in 1X PBS before mounting and imaging at 100X on an Olympus IX83 inverted confocal microscope with a microlens-based super-resolution imaging system (VT-iSIM; VisiTech International).

For the immunolabeling of Ybx1 in wild-type, heterozygous, and homozygous *ybx1* mutant larvae, we fixed 5 dpf zebrafish trunks in 4% formaldehyde for 3–4 h at room temperature, then rinsed with methanol prior to storage at −20 °C. Before immunostaining, we gradually rehydrated the trunks in fresh PBS, permeabilized them for 5 m with cold acetone (−20 °C), then washed them several times in 0.1% PBST. We incubated the trunks at room temperature for 2–4 h or overnight at 4 °C in a 0.1% PBST blocking solution containing 5% normal donkey serum and 2% DMSO, then again overnight at 4 °C in fresh blocking solution with rabbit anti-YB-1 (C-terminal) antibodies (1:100; Sigma-Aldrich, St. Louis, USA). After several washes in 0.1% PBST, we again incubated the trunks for 3 h at room temperature in fresh blocking solution with donkey Alexa Fluor 488 anti-rabbit antibodies (1:500; Invitrogen). After further washes in 0.1% PBST, we incubated the trunks for 10 m in DAPI (1:500) and stored them at 4 °C until imaging them at 60X magnification. When immunostaining *Tg(atoh1a:dTomato)* larvae, we additionally used monoclonal mouse anti-RFP antibodies (1:1000; Invitrogen) and donkey Alexa Fluor 561 anti-mouse antibodies (1:500; Invitrogen) during the primary and secondary incubations, respectively.

### Quantification and statistical analyses

For the quantification of HC and nonsensory cell counts across endpoint and time-course experiments, each data point $n$ represented a single NM and replicates $N$ represented individual larvae. In general, these counts corresponded to the first-primordium derived anteroposterior NMs of the posterior lateral line from the second to the fifth NM, though sometimes including the first, sixth, or terminal NMs. For the quantification of Ybx1 immunofluorescence in $atoh1a^+$ cells, data points $n$ represented single cells. Wherever present, the reported average values indicated the arithmetic mean over the pooled data points $n$ and the error bars indicated the standard error of the mean. The statistical significance for the mean cell-count differences—as well as for Ybx1 immunofluorescence—was computed using one-sided or two-sided $T$-tests for HCs

and $atoh1a^+$ cells or nonsensory cells, respectively, and the statistical power was computed using Cohen's $d$ as the effect size. For the quantification of NMs with $\geq 1$ new HC, the statistical significance was computed using a one-sided Fisher's exact test for proportions.

## RESULTS

### Supervised inference of the early GRN in regenerating central SCs and HCs

To reconstruct lateral-line NMs' dynamic GRNs, we fine-tuned DELAY on published scRNA-seq data of HCs and SCs immediately following injury, from zero to ten hours post ablation (hpa) of HCs (*Baek et al., 2022*). As separate ground truth, we also used published targets of *sox2*, *sox21a*, *six1a*, and *six1b* from single-cell ATAC-seq of inner-ear HC regeneration (*Jimenez et al., 2022*). To gather the original scRNA-seq of early NM regeneration, (*Baek et al., 2022*) sequenced the transcriptomes of single cells belonging to several sensory and non-sensory-cell types, including HC progenitors, young HCs, mature HCs, central SCs, dorsoventral (DV) SCs, anteroposterior (AP) SCs, amplifying SCs and mantle cells (Fig. 1A). From these cell types, we subsequently subset two trajectories that corresponded to central SC regeneration (0–10 hpa) and HC regeneration—consisting of central SCs (0–3 hpa), HC progenitors (1–5 hpa), and young and mature HCs (3–10 hpa)—which we used to infer trajectory-specific pseudotime values (Fig. 1B).

We trained DELAY on these trajectories and inferred the unique HC- and SC-specific GRNs for 303 TFs expressed in either lineage. To highlight the most important TFs across both GRNs, we combined DELAY's maximum predictions per TF-target pair and selected the top twenty regulators for each gene, which resulted in a combined GRN showcasing the key TFs and interactions coordinating NMs' early injury response and HC and SC specification (Figs. 1C, 1D). The central hub in the combined network, *Y-box binding protein 1* (*ybx1*), is a predicted regulator of 255 TFs involved in the regeneration of HCs and central SCs. Although no HC- or NM-specific roles are currently known, Ybx1 is a broadly expressed DNA- and RNA-binding protein that regulates transcription, translation, and cellular stress responses in other tissues (*Zasedateleva et al., 2002*; *Sun et al., 2018*; *Guarino et al., 2019*). *Baek et al. (2022)* detected *ybx1* mRNA in at least 50% of all NM cells, and especially in HC progenitors and young HCs, making it one of the highest-expressed TFs across the two trajectories (Fig. 1E).

### Ybx1 uses proximal bipartite motifs to regulate target genes' expression

Because DELAY relies primarily on highly expressing cells, strong mRNA expression provided a robust signal to identify *ybx1*'s putative targets (*Reagor, Velez-Angel & Hudspeth, 2023*). The algorithm's low false-positive rate also facilitated *de novo* discovery of enriched motifs across the enhancers of *ybx1*'s predicted targets. For each TF in the combined GRN, we gathered the enhancer sequences spanning 50 kb upstream and downstream of their transcription start sites (TSSs) and divided the sequences into 100 bp fragments, which we classified either as controls or as predicted *ybx1* targets. We then used STREME to perform discriminative motif analysis and identify enriched motifs in the predicted targets'

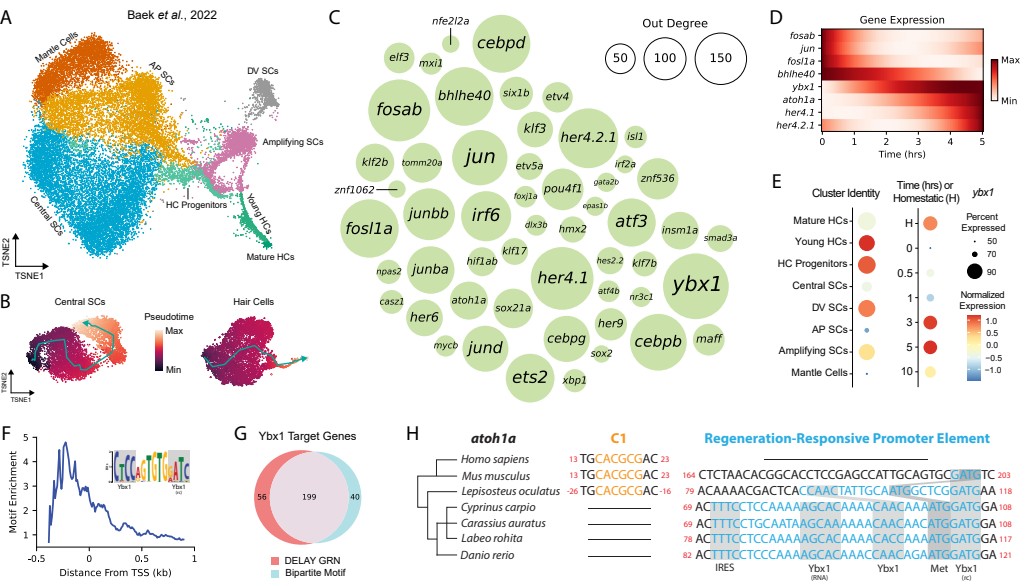

**Figure 1** **DELAY identifies *ybx1* as a global transcriptional regulator upstream of *atoh1a*.** (A) A *t*-SNE embedding depicts scRNA-seq data from *Baek et al. (2022)* containing HC progenitors, young HCs, mature HCs, central SCs, dorsoventral (DV) SCs, anteroposterior (AP) SCs, amplifying SCs and mantle cells during homeostasis and 0, 0. 5, 1, 3, 5, and 10 hpa. (B) Two pseudotime trajectories depict central SC and HC regeneration from 0-10 hpa. (C) A bubble plot shows key TFs in the combined DELAY GRN for HC and central SC regeneration. Bubble size represents outdegree centrality. (D) The expression of key TFs is depicted using generalized additive models fitted to cells' timepoints from 0–5 hpa. The expression of *ybx1* precedes that of *atoh1a*. (E) A dot plot shows that *ybx1* is expressed across all cell types and timepoints but is relatively enriched across HC progenitors and young HCs at 3–5 hpa. The analysis in (A–E) is based on data from *Baek et al., (2022)*. (F) Relative enrichment of bipartite motifs suggests that Ybx1 regulates transcription of its targets proximally to their transcription start sites (TSSs). (G) Two-thirds of 303 TFs in the combined GRN are predicted targets of *ybx1* and possess ≥1 bipartite Ybx1 motif in their enhancers. (H) Alignment of *atoh1a* revealed a candidate regeneration-responsive promoter element with two Ybx1 DNA-binding sites and a 5′ UTR element with Ybx1 RNA-binding and internal ribosome entry (IRE) sites. Cyprinid species also lack proximal C-sites (C1) for Notch-mediated suppression of *atoh1a* transcription (*Abdolazimi, Stojanova & Segil, 2016*). rc, reverse complement.

enhancers that were similar to Ybx1's known DNA-binding motifs (*Bailey, 2021*). We previously used the same approach to identify enriched motifs whose *de novo* sequences closely resembled known DNA-binding motifs for TFs such as *Gfi1* and *Hey1* during murine inner-ear HC development (*Reagor, Velez-Angel & Hudspeth, 2023*).

Using its highly conserved cold-shock domain (CSD), Ybx1 can interact with a broad range of sequences commonly comprising the consensus motif 5′-CNNC-3′or related sequences (*Zasedateleva et al., 2002*). Ybx1 therefore possesses immense flexibility to recognize and regulate the expression of target genes whose regulatory sequences contain variants of this short motif. Moreover, Ybx1 binds to single-stranded DNA (ssDNA)—often near other ssDNA-binding factors, or itself—to melt double-stranded DNA and regulate the transcription of its targets (*Lasham et al., 2000*; *Zasedateleva et al., 2002*; *Lyabin, Eliseeva & Ovchinnikov, 2014*). Proximally to its predicted targets' TSSs in the combined HC and SC GRN, we discovered an enriched, bipartite motif that contained two sequences that

resembled Ybx1's forward and reverse-complement DNA-binding sites (Fig. 1F). We subsequently used FIMO to identify 239 TFs in the combined GRN whose enhancers or promoters contained at least one occurrence of the *de novo* motif, 199 of which were also predicted targets of *ybx1* in the combined GRN (Fig. 1G) (*Grant, Bailey & Noble, 2011*). Together, these results suggest that Ybx1 can regulate diverse targets in regenerating HCs and central SCs by recognizing bipartite motifs that contain both forward (5′-CNNC-3′) and reverse-complement (5′-GNNG-3′) DNA-binding motifs.

### A candidate regeneration-responsive promoter element for *atoh1a*

We noticed that one Ybx1 bipartite motif overlapped with the start codon of *atoh1a*, suggesting that *ybx1* operates directly upstream of HC specification. Alignment of the surrounding nucleotides revealed a candidate regeneration-responsive promoter element (cRRPE) in cyprinids—carps and minnows—that contained two variants of Ybx1's forward (5′-CAAC-3′) and reverse-complement (5′-GATG-3′) DNA-binding sites (Fig. 1H). The orthologous alignment of *atoh1a* across cyprinids including *Danio rerio*, *Cyprinus carpio* (common carp), *Carassius auratus* (goldfish), and *Labeo rohita* (rohu) as well as other species such as *Lepisosteus oculatus* (spotted gar), *Mus musculus* (mouse), and *Homo sapiens* (human) suggested an origin for the cRRPE no later than the last common ancestor of the Neopterygii (*i.e.,* carps, minnows, and gars), resulting from the emergence of a forward Ybx1 DNA-binding site near a preexisting reverse-complement site.

We also identified in the 5′-untranslated region (UTR) of *atoh1a* a conserved element with a Ybx1 RNA-binding site—the so-called "dorsal localization element" (DLE), that includes the sequence AGCAC followed by a short hairpin or stem-loop (CCA-$N_6$-TGG) (*Zaucker et al., 2018*). This observation suggests that the cRRPE might also have evolved to regulate *atoh1a*'s translation. Ybx1 regulates the expression of multiple Nodal pathway genes in four-cell-stage embryos using DLEs to sequester transcripts and co-regulate their translation (*Zaucker et al., 2018*). Our identification of an identical element in the 5′UTR of *atoh1a* suggests that Ybx1 similarly regulates *atoh1a*'s translation, especially during periods of cellular stress when Ybx1's expression is at its highest. We further identified an internal ribosome entry site (IRES) adjacent to *atoh1a*'s DLE-like element, which is unexpected for a monocistronic mRNA and might also support *atoh1a*'s translation during periods of cellular stress (*Lyabin, Eliseeva & Ovchinnikov, 2014*; *Weingarten-Gabbay et al., 2016*). Overall, our observations of multiple, conserved DNA- and RNA-binding sites suggests that—in addition to its putative role in global transcriptional regulation—Ybx1 specifically regulates *atoh1a*'s expression at multiple levels.

### *ybx1* haploinsufficiency impairs the regeneration of lateral-line HCs

To investigate the impact of *ybx1* on HC development and regeneration, we obtained *ybx1*[sa34489] mutant zebrafish larvae that possessed an early stop in the ninth codon of *ybx1*. Because *ybx1* does not possess an alternative start codon prior to the CSD (*i.e.,* amino acids 56–104), homozygous *ybx1*[sa34489] larvae should not contain functional Ybx1 proteins with nucleic acid-binding capabilities (*Kumari et al., 2013*). We inbred heterozygous *ybx1* mutant adults to obtain wild-type, heterozygous, and homozygous mutant larvae, which

we then treated with neomycin to ablate their lateral-line HCs (Fig. 2A). At five days post-fertilization (5 dpf), we did not observe a difference in the number of HCs between the NMs of untreated *ybx1* mutant larvae and those of their wild-type siblings ($p > 0.2$; Fig. 2B; Supplementary Tables). However, the NMs of both the heterozygous and homozygous *ybx1* mutant larvae regenerated ~1.5 fewer HCs by two days post ablation (2 dpa) than their wild-type siblings ($p < 0.02$; Fig. 2B; Supplementary Tables). Moreover, we did not observe a statistically significant difference in the number of DAPI$^+$ nonsensory cells between regenerating NMs from wild-type and *ybx1* mutant larvae ($p > 0.9$; Fig. 2C). These results suggest that *ybx1$^{sa34489}$* is a dominant mutation that specifically disrupts the regeneration of lateral-line HCs in both heterozygous and homozygous *ybx1* mutant larvae.

To confirm that our mutants possessed no or reduced expression of Ybx1, we used commercially available anti-Ybx1 antibodies to perform immunolabeling of NMs from wild-type, heterozygous, and homozygous *ybx1* mutant larvae. As expected, we did not observe any Ybx1 expression in NMs of homozygous *ybx1* mutants (Fig. 2D). For immunolabeling of heterozygous and wild-type larvae, we additionally bred our heterozygous *ybx1* adults with *Tg(atoh1a:dTomato)* zebrafish to differentiate between Ybx1 expression in nonsensory cells *versus* in HC progenitors and young HCs. These experiments revealed that Ybx1 was broadly expressed across the cytoplasm of HCs, nonsensory cells, and peridermal cells from both the wild-type and heterozygous *ybx1* mutant larvae (Fig. 2E). Strong expression in the SCs and peridermal cells resembled previous immunostaining in caudal fins of adult fish, where Ybx1 contributes to stress granule formation following injury (*Guarino et al., 2019*). However, our quantification of Ybx1's immunofluorescence in *atoh1a$^+$* cells confirmed that the HC progenitors and young HCs from NMs of heterozygous mutants expressed significantly less Ybx1 than their wild-type siblings ($p < 0.001$; Fig. 2F). This cell type-specific decrease in Ybx1 expression may therefore explain the observed haploinsufficiency of heterozygous *ybx1* mutant larvae during regeneration.

### *ybx1* promotes the rapid regeneration of lateral-line HCs following ablation

Because *ybx1* is the central hub in the combined GRN for early regeneration, we wondered whether mutants' small but consistent deficits in HC number indicated an early delay in regeneration. To test this hypothesis, we again bred our heterozygous *ybx1* mutants with *Tg(atoh1a:dTomato)* zebrafish to yield equal numbers of wild-type sibling and heterozygous *ybx1* mutant offspring that expressed dTomato in their newly regenerated HCs (Fig. 2G). Because *atoh1a* expression decreased after ~12 h in mature HCs, the expression of *atoh1a* was a convenient marker for identifying new HCs as well as the downstream impacts of perturbations to that gene's putative regulators such as *ybx1*. We therefore used these larvae to perform time-course observations and quantified new HCs in regenerating NMs at 12, 16, 20, and 24 hpa (Fig. 2H). These observations, we reasoned, might reveal whether wild-type and heterozygous *ybx1* mutant larvae possessed different rates of HC addition or induction timing post-ablation.

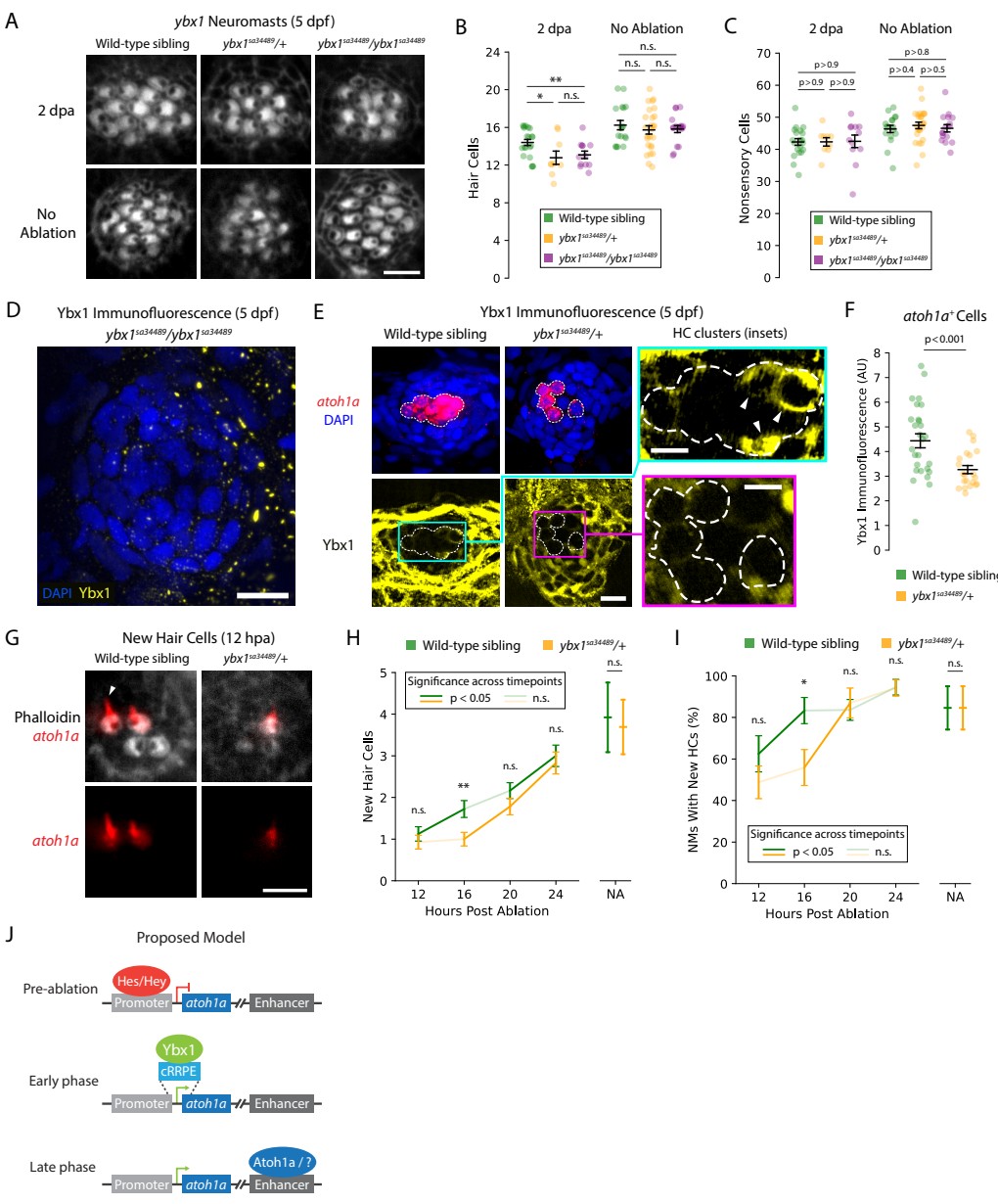

**Figure 2** ***ybx1*** **promotes rapid HC regeneration and *atoh1a* expression after damage.** (A) Representative images show phalloidin staining of HCs in post-ablation and control NMs from wild-type, heterozygous, and homozygous *ybx1* mutant larvae (scale bar = 4 μm). (B and C) Quantification of HCs and DAPI⁺ nonsensory cells from post-ablation and control NMs. Each data point represents a single NM. (D and E) Representative images of Ybx1 immunostaining across NMs from wild-type, heterozygous, and homozygous *ybx1* mutant larvae (scale bar = 10 μm). Inset images show Ybx1 accumulation (arrowheads) in *atoh1a*⁺ cells (inset scale bars = 5 μm). (F) Quantification of Ybx1 immunofluorescence in *atoh1a*⁺ cells from wild-type ($N = 4$) and heterozygous *ybx1* mutant ($N = 3$) larvae. Data points represent individual *atoh1a*⁺ HC progenitors or young HCs across wild-type ($n = 27$) and mutant ($n = 21$) categories. (continued on next page...)

**Figure 2 (…continued)**
(G) Representative fluorescence images show *atoh1a* expression in newly regenerated HCs from wild-type and heterozygous *ybx1* mutant larvae. dTomato marks HCs' apical surfaces and the growing kinocilia (arrowhead) in older HCs (scale bar = 4 μm). (H and I) Quantification of new HCs in control and 12–24 hpa NMs from wild-type and heterozygous *ybx1* mutant larvae. (J) Proposed model of *atoh1a* transcriptional regulation during pre-ablation, early-phase, and late-phase HC regeneration. Where shown, average quantities represent mean values across the pooled data points and the error bars represent SEMs. Complete descriptions of quantifications and statistical analyses can be found in the Methods section and Supplementary Tables (*, $p < 0.05$; **, $p < 0.01$).

As expected, we observed that heterozygous *ybx1* mutants exhibited delayed onset of HC regeneration following damage. Across mutant larvae, the number of new HCs per NM remained ∼1 between 12–16 hpa despite increasing to ∼1.75 in wild-type siblings ($p < 0.01$ at 16 hpa; Fig. 2H; Supplementary Tables). From 16–20 hpa, NMs from the heterozygous mutants belatedly upregulated their regeneration of new HCs, lagging their wild-type siblings by roughly 4 h ($p < 0.01$; Fig. 2H; Supplementary Tables). The proportion of wild-type NMs with ≥1 newly regenerated HC also led heterozygous mutants by approximately 4 h, reaching ∼80% by 16 hpa *versus* 20 hpa in mutants (Fig. 2I; Supplementary Tables). Interestingly, NMs from both wild-type and heterozygous larvae failed to recover the same rate of HC addition by 24 hpa as their untreated controls, indicating that normal development alone does not account for the delayed regeneration. Overall, these results demonstrated that *ybx1* lies upstream of *atoh1a* during early HC regeneration and promotes ∼20% faster regeneration in wild-type animals.

# DISCUSSION

Time-course scRNA-seq experiments such as those performed by (*Baek et al., 2022*) can uncover changing patterns of gene expression during dynamic processes such as lateral-line HC regeneration, ultimately facilitating *in silico* reconstruction of the underlying GRNs. Using our supervised deep-learning algorithm DELAY, we identified *ybx1* as the central hub in the combined GRN for early HC and central SC regeneration. Our computational and experimental results indicated that *ybx1* promotes the rapid regeneration of HCs in damaged NMs and suggested a new mechanism for early transcriptional control of *atoh1a*. In mice, *Atoh1* uses its 3′enhancer to upregulate its own expression in a self-driven switch from suppression to activation (*Helms et al., 2000*). However, Notch effectors such as Hes and Hey TFs can suppress *Atoh1* expression by binding to conserved sites such as C1 through C4 in *Atoh1*'s promoter as well as long-range enhancers for regeneration (*Zine et al., 2001*; *Abdolazimi, Stojanova & Segil, 2016*; *Shi et al., 2024*). Nevertheless, derepression of *Atoh1* by Notch effectors is insufficient to drive its initial expression—a phenomenon known as activator insufficiency (*Abdolazimi, Stojanova & Segil, 2016*). Our results here suggest that *ybx1* can initially upregulate *atoh1a*'s expression through interactions with the cRRPE (Fig. 2J), likely aided by the absence of the most proximal C-site (C1) in zebrafish and related cyprinids (*Abdolazimi, Stojanova & Segil, 2016*).

Our model can explain the haploinsufficiency of heterozygous *ybx1* mutant larvae as well. Theoretical and experimental investigations into other TFs' mechanisms of

positive autoregulation suggest that this class of regulatory interactions can generate sharp gene-expression thresholds, below which genes' expression will remain inactivated (*Alon, 2007*). The decreased abundance of Ybx1 in $atoh1a^+$ cells of heterozygous *ybx1* mutants might therefore reduce *atoh1a*'s expression below an evolutionarily tuned self-activation threshold. Further *in silico* experiments can investigate this hypothesis by performing stochastic simulations of the combined GRN using genotype-specific rates of *ybx1* transcription for wild-type, heterozygous, and homozygous mutant networks.

Further experiments are also necessary to separate Ybx1's role in transcriptional *versus* translation control of *atoh1a*. Our computational analysis with DELAY suggested that this interaction primarily occurs at the level of transcription, but the existence of conserved DLE-like and IRE sites in the cRRPE—as well as Ybx1's cytoplasmic localization in homeostatic $atoh1a^+$ cells—implied a further role in translation as well. Future studies can separate these activities by employing techniques such as CRISPR-Cas9 base editing to precisely alter individual DNA- and RNA-specific binding sites in the cRRPE. If successful, these experiments might also separate the role of Ybx1 in genome-wide *versus atoh1a*-specific transcriptional regulation. Alternative approaches such as ChIP-seq or cross-linking and immunoprecipitation with sequencing (CLIP-seq)—or specific approaches such as ChIP-qPCR or CLIP-qPCR—can confirm direct binding of Ybx1 to *atoh1a* or its transcripts, respectively.

Because the cRRPE exists only in carps and minnows, *ybx1*-mediated *atoh1a* upregulation is unlikely to extend to other species such as mice or humans. However, it will be interesting to determine whether *ybx1* can promote the rapid regeneration of inner-ear HCs and lateral-line HCs in adult zebrafish as well. Regardless, our results here strongly support the continued use of DELAY to reconstruct dynamic GRNs from genes' expression in single cells. Transfer learning can identify important patterns from noisy representations of dynamic processes such as pseudotime trajectories (*Reagor, Velez-Angel & Hudspeth, 2023*). Our results support DELAY's ability to identify GRN hubs whose dynamic expression propagates downstream to genes such as *atoh1a* or arises from complex upstream regulation such as *ESRRB* (*Mannens et al., 2024*). Future studies can use DELAY to elucidate causal mechanisms underlying genes' temporal regulation across diverse biological systems.

## ACKNOWLEDGEMENTS

We would like to thank Gaurav Shrestha, Nicolas Velez, Emily Atlas, Agnik Dasgupta, and other members of the Laboratory of Sensory Neuroscience for helpful discussions, as well as Samantha Campbell for expert zebrafish husbandry. We also thank Tatjana Piotrowski for kindly providing *Tg(atoh1a:dTomato)* larvae.

### Funding

Caleb C. Reagor is supported by the National Science Foundation Graduate Research Fellowship Grant No. 1946429. The funders had no role in study design, data collection and analysis, decision to publish, or preparation of the manuscript.

## Grant Disclosures

The following grant information was disclosed by the authors:
National Science Foundation Graduate Research Fellowship: 1946429.

## Competing Interests

The authors declare there are no competing interests.

## Author Contributions

- Caleb C. Reagor conceived and designed the experiments, performed the experiments, analyzed the data, prepared figures and/or tables, authored or reviewed drafts of the article, and approved the final draft.
- Paloma Bravo conceived and designed the experiments, performed the experiments, authored or reviewed drafts of the article, and approved the final draft.
- A.J. Hudspeth conceived and designed the experiments, authored or reviewed drafts of the article, and approved the final draft.

## Animal Ethics

The following information was supplied relating to ethical approvals (i.e., approving body and any reference numbers):

Experiments were performed in accordance with the standards of Rockefeller University's Institutional Animal Care and Use Committee with protocol number 22,081 H (PRV 19,076 H).

## Data Availability

Data from *Baek et al. (2022)* is available at the Gene Expression Omnibus: GSE196211.

Preprocessed data from *Baek et al. (2022)* and from this study are also available at

– Github and Zenodo: https://github.com/calebclayreagor/ybx1-HC-regeneration.

Caleb C. Reagor. (2025). calebclayreagor/ybx1-HC-regeneration (v0.2).

– Zenodo. https://doi.org/10.5281/zenodo.15525631.

Raw fluorescent microscopy images are available at Dryad:

– Development and regeneration of inbred *ybx1* mutant larvae (2025).

[Dataset]. Dryad. https://doi.org/10.5061/dryad.v15dv4257.

– Time-course regeneration of wild-type larvae (2025).

[Dataset]. Dryad. https://doi.org/10.5061/dryad.g1jwstr1z.

– Time-course regeneration of *ybx1* heterozygous mutant larvae (2025).

[Dataset]. Dryad. https://doi.org/10.5061/dryad.jh9w0vtn5.

– *Ybx1* immunostaining (2025).

[Dataset]. Dryad. https://doi.org/10.5061/dryad.wm37pvn0h.

## Supplemental Information

Supplemental information for this article can be found online at http://dx.doi.org/10.7717/peerj.19949#supplemental-information.

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
