# Peer review of "ybx1 acts upstream of atoh1a to promote the rapid regeneration of hair cells in zebrafish lateral-line neuromasts"

_PeerJ, doi:10.7717/peerj.19949_

## Round 0.1 · original submission · Major Revisions

As you can see, the three reviewers were enthusiastic about the potential for this work. However, they all raised concerns that will need to be addressed prior to publication. All reviewers feel that the paper is generally well written, but that the readability needs to be improved to increase its accessibility. Some of the changes they suggest will help with this. However, I would suggest a thorough proof read again to explore if the writing can be improved.

Reviewer 1 ·

Basic reporting

This manuscript describes computational analysis of a previously published RNA-seq data set to identify Ybx1 as candidate for a novel regulator of hair cell regeneration in the zebrafish lateral line. Further analysis revealed conserved Ybx1 binding motifs in putative target genes near transcription start sites. Analysis of atoh1a showed several Ybx1 motifs were associated with a "candidate regeneration-responsive promoter element" that is conserved amongst several cyprinid fish species. The authors also analyze Ybx1 mutants and find that hair cell regeneration is delayed. Overall, the paper documents the practical application of their DELAY platform to analyze transcriptomic data sets and identify important genetic "hubs" that might otherwise be overlooked. The paper is relatively short but reasonably sound in its execution and interpretation. Writing is adequate, but is terse in some places, making it less accessible to the general reader.

Specific comments:

1) The legend for Figure 1 needs more detail. Please explain how to interpret Fig. 1A, e.g. the meaning of the shading and size of the squares and the numbers inside each. The reader should not have to deduce this information. In Fig. 1B, explain what the label on the X axis means. In Fig. 1D, explain which targets are being analyzed. In Fig. 1E, explain what is meant by "repressive C1 site". This is briefly mentioned in the Results section, but the significance of its loss in cyprinids should be made explicit.

2) Most readers will not be familiar with Ybx1, so it could be confusing that the binding motifs identified in Figure 1 do not appear similar to each other. Please provide more context and explanation in the text about the diverse array of Ybx1 binding sites and how they have been identified.

3) The first paragraph on the second page of the Results ends with the sentence "The IRE and RNA-binding sites suggest that Ybx1 can stimulate cap-independent translation of atoh1a mRNA while suppressing global, cap-dependent translation". The rationale for this conclusion is mysterious and requires reading of the cited review article. This sentence may be better placed in the Discussion since it is speculative and is based on unpublished observations made by the authors of the review article. It should also be explained a more fully in order for it to make logical sense.

4) The second sentence of the Introduction states that the lateral line is derived from the otic placode, but it should state it is derived from a cranial placode.

Experimental design

This is fine.

Validity of the findings

This is fine.

Reviewer 2 ·

Basic reporting

This is a nicely written manuscript that utilizes a new machine learning algorithm to identify potentially important transcription factors that may control hair cell regeneration and validates one of the top hits by analyzing the effect of a loss-of-function mutation.
Overall, the writing is compact and appropriate for the paper, though there are a few places where additional text in the body of the manuscript could be helpful.
For example, the introduction might benefit from a little more discussion about the intention of using the DELAY, and the basis for this particular program (which is only briefly discussed in the beginning of the results). Within either this place or the results, a little more information about the specific ways in which this particular network might be expected to provide new information about expression datasets could be helpful context that seems barely discussed in setting up the key element of the paper.
There are a couple of places where I might suggest some additional textual editing:
1) The first sentence reads confusingly as humans do not have a lateral line system. It feels like there is a little too much compressed into that sentence.
2) On line 31, I would strongly suggest that supporting cells be defined as a distinct cell type and then secondarily that they can serve as HC progenitors but that is not necessarily the sole or primary function.
3) On line 36, it is not clear what “regenerate HC progenitors within 3-5 hours” means. Is this meaning the time for the SCs to initiate the regenerative program? Since most of the focus of this paper is on the SC->HC regeneration, I don’t think this is meaning the SC->2XSC regeneration of SCs that happens peripherally, but that’s more what the sentence implies.
4) Line 39-40, the discussion of SCs is again a little confusing here as there is an “instead” but not really a sense of what that is instead of.
References seem mostly appropriate for their respective sections.

Experimental design

Experimental design:
Overall the design seems appropriate, with a few suggestions/comments below.
From a read of the results section (line 55) I thought that this paper was re-analyzing the data from Baek et al., but the methods section almost implies that new sequencing was done.
More importantly, the description in the results section of this analysis (line 54) suggests a comparison between HCs and SCs, while the original study classified several different cell states (some of which are used later in this paper). It might be beneficial to discuss in further detail what subcategorization was used for the novel analysis.
Similarly, I think there could be more discussion of the GRNs “in either lineage” (line 56) particularly given that there are important distinctions between SCs, SCs becoming HC progenitors, proliferating cells, young HCs, and mature HCs.
It might be helpful to officially say that Figure 1B is a pseudotime analysis (I think?) since the data used does not have that sort of resolution.
On line 73, there is discussion about an overlap with the start codon of Atoh1, which seems a somewhat unusual spot for transcriptional regulation (since it is discussed primarily as a transcription factor); it is later discussed potentially as a translational regulator, but this isn’t elaborated on in the discussion, where it is solely discussed as a transcription factor.
The discussion of the data in Figure 1E is very limited in the results section or the discussion. It’s not clear what an IRES site upstream of the start codon is doing in a monocistronic RNA, nor is the numbering or relationship between the C1 and the RRPE clear.
For Figure 2, one important concern is the use of 3dpf fish, as this is very young where hair cells are still developing in most of the neuromasts and the HCs are less sensitive to damage (see Murakami et al., 2003 who found greatly reduced sensitivity at 4dpf and Baek et al. used 5 dpf). Additionally, no information is provided in the methods or results about how counts led to the graphs presented (based on looking at the data presented on the github, I’m guessing that a few NMs per fish – though not clear which ones as it’s certainly not all – were averaged to create one “average” per fish which is plotted and used for stats?). Additional information about that stats used should also be provided in the methods (as there is none – and I’m not sure why a U-test rather than a T-test is used here).
On the graphs of Figure 2 it isn’t clear why some lines are light colored and others darker.
Perhaps most importantly, the paper hinges on a claim that ybx1 is regulating atoh1a, but there is at best a slight delay in doubling of the HCs, but there is atoh1 expression even at 12h, so I’m wondering if this is more a delay in proliferation rather than control of atoh1a (or whether there might be differences that more SCs are becoming SC progenitors instead of HCs?).
For the discussion, I find Figure 2F presented without much information either in the text or in the legend. I think the legend also needs to clearly define this as a putative model as there is little data to back up how this is happening.
The discussion overall is very short, and the discussion of DELAY takes up most of it. I guess it would be nice to understand more clearly how the particular results of this paper clearly demonstrate the usefulness of this analysis.

Validity of the findings

See above for a few important comments on findings and how they are discussed.

·

Basic reporting

In the manuscript, “ybx1 acts upstream of atoh1a to promote the rapid regeneration of hair cells in zebrafish lateral-line neuromasts” (#110667) by Caleb C. Reagor and AJ Hudspeth falls within the scope of PeerJ. On publicly available data, the authors use a supervised learning algorithm, DELAY, to identify gene regulatory networks involved in lateral line hair cell regeneration. Using DELAY, the authors identify a key transcription factor, ybx1, predicted to play roles in lateral line hair cell regeneration. Using computational analysis, the authors predict a ybx1 motif associated with the atoh1a promoter and characterize the role of ybx1 in lateral line hair cell regeneration.

The research topic, methodology, and findings are fascinating and have great potential to advance the field of zebrafish hair cell regeneration. Overall, the manuscript's readability can be revised to improve the flow of the text, but the language is clear and professional throughout. The literature references are appropriate and there is enough background provided. While the hypotheses are clearly presented, there is insufficient information presented for readers to follow the procedures and results in the figures provided. The figure legends and methodology do not provide enough information to support the authors conclusions, such as information about the statistical analysis and genotypes.

Experimental design

The research question is well-defined and very meaningful.
Some aspects of the investigation need more detail and information.

Validity of the findings

Although asterisks are provided suggesting stastics, the authors must include p values in the legend and also include how they performed the statistical analysis in the methods section. Controls are not well defined in the text although "Sibling" is indicated in the figures. Numbers of replicates are also not specified. Figures also do not reflect the text. Please see comments in section 4.

Additional comments

Recommendations to strengthen the manuscript are as follows:

1. Further characterization of the heterozygous ybx1sa34489/+ and homozygous ybx1sa34489/ sa34489 mutants needed. The ybx1sa34489 mutation contains an early stop in the 9th codon.
a) Is the protein non-functional in homozygous animals?
b) Are expression levels of ybx1 altered in heterozygous animals compared to wild-type controls? Do either heterozygous (ybx1sa34489/+) or homozygous (ybx1sa34489/ sa34489) have normal supporting cell numbers compared to wild-type siblings?
2. Ybx1 antibodies to detect the protein are commercially available. The authors should determine if the full-length protein is localizing to supporting or hair cells in the lateral line in homozygous mutants vs heterozygous mutants compared to wild-type controls. If Ybx1 protein is not localized or not present in the lateral line of homozygous animals, then this would support the hypothesis that the absence of Ybx1 during regeneration alters Atoh1a regulation.

3. The authors indicate in the methods section that they used Seurat to cluster previously published data and then conducted pseudotime trajectory analysis on this data. Including the clustering and pseudotime inference in Figure 1 will complement the data in Figure 1BC and strengthen the conclusions.

4. According to the materials and methods section, the authors computationally predict ybx1 direct targets in the zebrafish genome. The manuscript can be strengthened by explaining the results of the motif analysis in the text and showing a panel of these results (# of genes with motifs in the zebrafish genome vs # of differential genes with motifs in the genome) in a Figure or panel for Figure 1.
a) How many predicted direct targets are there in the zebrafish genome?
b) In addition to atoh1a, How many predicted direct targets are associated with differentially expressed genes from previously generated scRNA-seq data on the regenerating lateral line (or regenerating zebrafish adult inner ear)?
This information will strengthen the authors hypothesis that atoh1a is a putative direct target of ybx1 during regeneration.

5. Line 74- 87: The authors describe alignments across multiple species. This is very interesting data since regeneration competent species are likely to share conserved regulatory regions.
a) The conservation analysis methodology should be written in the materials and methods section even if just a simple alignment.
b) In addition, the annotations used for each species should be indicated in the methods section.

6. Figure 1A –
1) It is unclear what the numbers in figure 1A mean. If the image includes the numbers, the figure legend must indicate in writing what these numbers mean, especially since there is no indication of what the numbers mean in the text.
2) DELAY nodes have been previously published using Circles according to the citation referenced. The current visual depiction of the DELAY nodes is acceptable but difficult to interpret with the gray scale and squares. Are there other ways to present the DELAY nodes in a more aesthetic manner?
3) Is the candidate regeneration responsive promoter enhancer also conserved in zebrafish? It is unclear from the alignments and text, if the enhancer is conserved across species including the zebrafish.

7. Figure 2
1) Figure 2A In the text, line 94, the authors inbred hets to obtain wild-type siblings, hets and homozygous mutant larvae.
a) In Figure 2A - Is the mutant either heterozygous (ybx1sa34489/+) or homozygous (ybx1sa34489/ sa34489) and why cant data from heterozygous and homozygous mutant animals be shown? Sibling should be “wild-type sibling”.
b) What does Figure 2A show? I can see some difference between wild-type sibling and mutant. However this observation can be supported via quantification.
i) In addition, it is unclear what is being shown: Is this a representative neuromast of what the authors were scoring? Which neuromast is this image of?

8. Figure 2B
1) Data from wild-type siblings, heterozygous, and homozygous larva should be presented. At the moment only wild-type siblings and “mutant” data is shown.
2) Is the mutant heterozygous or homozygous? The corresponding genotype needs to be indicated and data from heterozygous and homozygous mutants will support the conclusion that the ybx1sa34489/+ mutation is dominant.

9. Figure 2D and 2F
1) Hair cell counts for Non-regenerating control should be present.
2) Since the hair cell ablation studies are being performed at 3 dpf, the lateral line is still developing. By 24hours post ablation, it looks like hair cell regeneration catches up – How do you account for regeneration vs normal growth?
3) Is zebrafish inner ear hair cell regeneration impacted in the ybx1sa34489 mutants?

10. Figure 2C –
1) What is this data supposed to indicate? Are images from animals that are wild-type or heterozygous or homozygous? Are these mutant or wildtype? The figure can be more informative if images from wild-type siblings, hets and homozygous mutant larvae are shown and labeled.

11. Materials and Methods comments:
a) Information about how the authors performed their statistical analysis on hair cell regeneration data (related to figures 2B, 2D, 2F) should be described in the materials and methods section. The authors should describe how they generated error (e.g. standard deviation) and the p value’s need to be presented in the text or figure legend.
b) There is no information in the text or methods section on how hair cell averages were calculated.
c) Which neuromasts were examined? Are the averages from four Neuromasts or just one Neuromast per fish?

12. Comment
Discussion of Future directions may include:
1) mutating the cRRPE and demonstrating that a mutation also alters atoh1a expression and also contributes to a hair cell regeneration phenotype,
2) or ChIP-qPCR with ybx1 antibody, and qPCR the atoh1a locus to confirm direct binding.
3) Determining if ybx1sa34489 impacts inner ear regeneration in larval or adult contexts

---

## Round 0.2 · Minor Revisions

Three reviewers have now evaluated the resubmitted manuscript, and all are in agreement that it is vastly improved. As you will see, one reviewer requests some minor corrections to make it ready for publication.

Reviewer 1 ·

Basic reporting

No comment.

Experimental design

No comment.

Validity of the findings

No comment.

Additional comments

This revised manuscript addresses all of my previous concerns and is much improved. It now presents a linear and compelling account of the authors' identification and analysis of ybx1 function during regeneration of lateral line neuromasts in zebrafish. Figure 1, in particular, is far more intuitive and clear. Additionally, the anti-Ybx1 staining in Figure 2 is also helpful in validating the mutant as well as revealing where ybx1 is expressed - the images are clear and persuasive. Overall, this is a concise, well written paper that provides excellent support for a novel regulator of hair cell regeneration.

Reviewer 2 ·

Basic reporting

This manuscript has been significantly improved by the revisions and updates to the text and figures. There are a few additional places where some clarity could be provided:
1) In the updated first sentence, I’m not sure Corwin (1981) is the best sole reference for that; while it is the first note of postnatal production of hair cells, it’s limited to sharks and there have been several other papers that have examined fish and amphibian regeneration in the inner ear that would be important to cite there.
2) In lines 50/51, I think the discussion of the central supporting cells could be improved a bit. In general, I think there is some confusion of language across different labs talking about the same cells/processes but with varying words and definitions. In particular here, it might be good to clarify that some of the central supporting cells upregulate expression of atoh1a following damage (expression at baseline is quite low in these cells) which pushes them to divide symmetrically into hair cells.
3) In line 82, I’m not sure I understand the Hao reference, which is not the original Baek study and comes after it, so that can’t be the methods used in Baek.
4) Line 147: Hair cells at 3dpf are not more susceptible to complete ablation, and indeed are somewhat less sensitive to ablation because the hair cells are younger and have less developed transduction apparatus (see Murakami et al., 2003 among other papers). Baek (and most other literature) uses 5dpf for this reason. Not going to insist that your work be redone at 5dpf, but there should be clarity in this area.
5) In the area around lines 292-298, I greatly appreciate the new figure showing Ybx1 expression, however it feels like the discussion of this figure under-addresses two important features. First, although the text states that the protein is broadly expressed in HCs, nonsensory cells and peridermal cells, it is clear from these pictures that the HC expression is the lowest. It might be worth some commentary on the strong expression of this in the supporting and other cells (and/or seeing whether expression changes in response to neomycin). The second feature shown in the pictures is that the protein seems to be exclusively cytoplasmic in all of the cell types (HC and SC certainly) with nuclei notably lacking the protein. This has important implications for the function of the protein, suggesting that the RNA-binding/translational control may be far more significant a role than the transcriptional regulation (which would require nuclear protein). It seems this is worth noting here and adding to the discussion, particularly around lines following 349 in discussing the transcriptional/translational role.

Experimental design

The concerns #4 and #5 above cross over a little into this space, but I have no additional experimental concerns for this short report.

Validity of the findings

These data seem valid and appropriate. Additional experimentation could take it further but for this scope of this report everything is adequeate.

Additional comments

None

·

Basic reporting

I have a minor comment for Figure 2: Include a scale bar for panels 2A, 2D, 2G.

Experimental design

No comment.

Validity of the findings

No comment.

Additional comments

In the manuscript, “ybx1 acts upstream of atoh1a to promote the rapid regeneration of hair cells in zebrafish lateral-line neuromasts” (#110667) by Caleb C. Reagor, Paloma Bravo, and AJ Hudspeth falls within the scope of PeerJ. The authors use a supervised learning algorithm, called DELAY, to identify gene regulatory networks involved in lateral line hair cell regeneration. Using Delay, the authors identify a key transcription factor, ybx1, predicted to play roles in lateral line hair cell regeneration. Using computational analysis, the authors predict a ybx1 motif associated with the atoh1a promoter and characterize the role of ybx1 in lateral line hair cell regeneration.

The research topic, methodology, and findings are fascinating and advance the field of zebrafish hair cell regeneration. In the revised manuscript, the authors significantly improved the readability and flow of the text. The hypotheses are clearly stated, and the authors have included sufficient information for readers to follow the procedures and interpret the results. The figures have been improved, and the data appear robust. The authors addressed my concerns related to basic reporting, experimental design, and validity of the findings, and I recommend the revised manuscript for publication.

---

## Round 0.3 · accepted · Accept

Thank you for your revisions; the paper is now suitable for publication. Thank you for considering PeerJ for this excellent work - it will make an exciting contribution to the field.